# Evaluation of the Stability of a 1,8-Cineole Nanoemulsion and Its Fumigant Toxicity Effect against the Pests *Tetranychus urticae, Rhopalosiphum maidis* and *Bemisia tabaci*

**DOI:** 10.3390/insects14070663

**Published:** 2023-07-24

**Authors:** Rocío Ayllón-Gutiérrez, Eduardo Alberto López-Maldonado, Mariana Macías-Alonso, Joaquín González Marrero, Laura Díaz-Rubio, Iván Córdova-Guerrero

**Affiliations:** 1Facultad de Ciencias Químicas e Ingeniería, Universidad Autónoma de Baja California, Tijuana 22390, Mexico; ayllonr@uabc.edu.mx (R.A.-G.); elopez92@uabc.edu.mx (E.A.L.-M.); 2Instituto Politécnico Nacional, Unidad Profesional Interdisciplinaria de Ingeniería Campus Guanajuato, Av. Mineral de Valenciana 200 Col. Fracc. Industrial Puerto Interior, Silao 36275, Mexico; mmacias@ipn.mx (M.M.-A.); jgonzalezm@ipn.mx (J.G.M.)

**Keywords:** 1,8-cineole, 1,3,3-trimethyl-2-oxabicyclo[2.2.2]octane, eucalyptol, nanoemulsion, crop pests, essential oils, *Tetranychus urticae*, spider mite, *Bemisia tabaci*, silverleaf whitefly, *Rhopalosiphum maidis*, corn leaf aphid, pesticides

## Abstract

**Simple Summary:**

Pests are largely responsible for the loss of agricultural crops, so the search for new pesticides for their control is indispensable. The irrational use of synthetic products has affected human health and the environment, and has generated resistance, which is why the use of natural products could be a safer alternative. In this study, the formulation of a nanoemulsion was carried out to provide greater stability to the botanical compound 1,8-cineole, whose pesticidal effect is documented; however, it is not used due to its high volatility. The pesticidal effect of this nanoemulsion was also evaluated against the pests known as two-spotted spider mite, corn aphid and silverleaf whitefly. Nanotransport systems such as nanoemulsions allows for the field application of molecules that are chemically unstable on their own, such as monoterpenoids. With this study it was possible to determine that these nanoemulsion systems favor the release of the active compound in laboratory tests, increasing the mortality rate of the three pests. This allows for proposing this nanoemulsion as a potential botanical pesticide product against agricultural arthropod pests.

**Abstract:**

Pest control is a main concern in agriculture. Indiscriminate application of synthetic pesticides has caused negative impacts leading to the rapid development of resistance in arthropod pests. Plant secondary metabolites have been proposed as a safer alternative to conventional pesticides. Monoterpenoids have reported bioactivities against important pests; however, due to their high volatility, low water solubility and chemical instability, the application of these compounds has been limited. Nanosystems represent a potential vehicle for the broad application of monoterpenoids. In this study, an 1,8-cineole nanoemulsion was prepared by the low energy method of phase inversion, characterization of droplet size distribution and polydispersity index (PDI) was carried out by dynamic light scattering and stability was evaluated by centrifugation and Turbiscan analysis. Fumigant bioactivity was evaluated against *Tetranychus urticae*, *Rhopalosiphum maidis* and *Bemisia tabaci*. A nanoemulsion with oil:surfactant:water ratio of 0.5:1:8.5 had a droplet size of 14.7 nm and PDI of 0.178. Formulation was stable after centrifugation and the Turbiscan analysis showed no particle migration and a delta backscattering of ±1%. Nanoemulsion exhibited around 50% more bioactivity as a fumigant on arthropods when compared to free monoterpenoid. These results suggest that nanoformulations can provide volatile compounds of protection against volatilization, improving their bioactivity.

## 1. Introduction

One of the main problems in the agriculture industry is the impact of crop pests and diseases on the yield of commercial crops. At this time, it is calculated that nearly 40% of global crop yield is being negatively affected by insect pests and diseases [1]. Some insects that are deemed destructive agricultural pests with multiple host plants are the two-spotted spider mite, the corn leaf aphid and the silverleaf whitefly.

Two-spotted spider mite *Tetranychus urticae* Koch (Acari: Tetranychidae) is a pest with cosmopolitan distribution that affects a wide range of ornamental, fruit and vegetable crops of economic significance [2,3,4]. These mites live on the underside of leaves, piercing the mesophyll cells with its stylet to feed, causing the characteristic chlorotic spots [5,6]. Control of *T. urticae* is heavily reliant on synthetic pesticides and acaricides; this, however, has led to a rising resistance [7,8,9,10,11].

*Rhopalosiphum maidis* Fitch (Hemiptera: Aphididae)*,* also known as corn leaf aphid, is one of the most common pests of maize, as well as other cereals such as barley, oats, rye, millet and sorghum crops around the world [12], with average annual losses of 10–15% of corn for human consumption [13]. Damage to crops is caused through direct feeding on the sap, by mold growing on its honeydew, or by vectoring of disease-causing viruses [14,15], including the maize dwarf mosaic virus (MDMV), barley yellow dwarf virus, sugarcane yellow leaf virus, potato virus Y, and maize chlorotic dwarf virus [16,17].

*Bemisia tabaci* (Hemiptera: Aleyrodidae), commonly known as silverleaf whitefly, comprises at least 44 cryptic species that are morphologically indistinguishable [18], and is considered one of the most destructive agricultural pests with more than 600 host plant species around the world [19], causing major losses to commercial crops including sweet potato, tomatoes and cucurbits [20,21,22]. Both nymphs and adults feed on the plants phloem and excrete honeydew which can lead to black mold growth on the leaves [16,23]. In addition, *B. tabaci* can transmit at least 100 different plant viruses including tomato yellow leaf curl virus (TYLCV) [24,25,26].

The fast development rate of these three pests favors a quick rise in the population causing a rapid decline in host plants [19,27,28]; the role of *R. maidis* and *B. tabaci* in the transmission of plant disease causing virus as well as mold growth, and the indiscriminate and abusive use of synthetic pesticide that benefits the rapid rising in resistance, such as the reported for *B. tabaci* and *T. urticae,* make their control a priority for farmers [20,29,30,31,32,33,34].

In recent years, there has been a growing concern regarding the negative impact of unreasonable application of synthetic pesticides on both human health and the environment. Secondary metabolites from plants have been proposed as a safer, effective and more sustainable alternative to conventional pest management approaches. Several phytochemicals have reported bioactivities against insects; among them, monoterpenoids, the main constituents of essential oils found in many aromatic plants, are part of the plants’ natural chemical defense against pathogens, insects and other herbivores [35].

Both essential oils and their monoterpenoids have demonstrated effectiveness against a variety of arthropod pests of relevance [36,37,38,39,40,41], including *T. urticae* and other phytophagous acari [42,43,44,45,46,47,48,49,50].

1,8-Cineole, also called eucalyptol, is a widespread monoterpene found in numerous aromatic plant essential oils (EOs), including eucalyptus and sage oils. This monoterpene has been reported as a natural insecticide and repellent compound [50,51,52,53]. However, in realistic settings, EOs and monoterpenes are not as effective due to their high volatility, low water solubility and chemical instability [54]. Nanoemulsions are suitable bioactive compound carriers for volatile oils, with their smaller droplet size compared to bulk emulsion making them less propense to instability processes [55]; as a result, this allows the retaining of volatile compounds for longer periods, extending their effectiveness [56].

This study aims to assess the efficiency of a nanoemulsion of monoterpene 1,8-Cineole comparing free and semi-hermetic condition 1,8-Cineole against *T. urticae*, *R. maidis* and *B. tabaci* in in vitro assays. Nanoemulsions are proposed, in this work, as a potential solution that allows a practical application of these compounds, minimizing the volatilization of the monoterpenic compound, thus maintaining its biocidal capacity. This research provides a rationale for further studies aiming to develop botanical-based pesticides.

## 2. Materials and Methods

Natural Eucalyptol (1,8-Cineole) ≥99% was purchased from SAFC^®^ Sigma-Aldrich. Tween 20 was obtained from Sigma-Aldrich (Merck México, Naucalpan de Juárez, Mexico), and Triton X-100 was acquired from Productos Químicos Monterrey (Productos Químicos Monterrey, S. A. de C. V., Monterrey, Mexico).

### 2.1. Formulation of Nanoemulsion

Emulsions were formed using the low energy method of emulsion phase inversion (EPI), which consists in the slow addition of an aqueous phase into an organic phase under constant stirring [57]. The final formulation was prepared by combining the 1,8-Cineole and a binary mixture of non-ionic surfactants Triton X-100 and Tween 20 (7:3), under magnetic stirring for 30 min. Deionized water was titrated at 1.5 mL/min flow, under constant stirring. Additional magnetic stirring was performed for 2 h to allow the formation of the nanoemulsion. Final composition of nanoemulsion was 5% oil (*w*/*w*), 10% total surfactant (*w*/*w*), and 85% water (*w*/*w*). The formulation was stored at room temperature (25 ± 2 °C) and characterized after 24 h. An additional emulsion with an oil: surfactant: water ratio of 0.5:0.5:9 was made to observe the influence of surfactant content in stability against centrifugal force.

### 2.2. Characterization of Nanoemulsion

Nanoemulsion droplet size distribution and polydispersity index (PDI) were measured by dynamic light scattering (DLS) (ZetasizerNano ZS-90, Malvern Instrument, Malvern, UK). No previous treatment was performed prior analysis.

To determine stability, the formulated emulsions were subjected to gravitational stress test by centrifugation at 3500 rpm for 30 min, then carefully observed for any phase separation.

Stable formulations were further analyzed on a Turbiscan LAB Stability Analyzer (Formulaction Scientific Instruments, Toulouse, France), using three kinds of data processing methods, transmitted intensity (T), backscattering intensity (BS) and Turbiscan stability index (TSI), to analyze the destabilization process of the nanoemulsion.

### 2.3. Acaricidal Activity

#### 2.3.1. Tested Animals

Tested acari *T. urticae* Koch were reared on bean plants (*Phaseolus vulgaris* L.) leaves in the Laboratory of Natural Products Chemistry, Faculty of Chemistry and Engineering, Autonomous University of Baja California. Rearing conditions were 26 ± 2 °C and 70 ± 5% R.H. with a 16:8 h (L:D) natural photoperiod. Mites were not previously exposed to any pesticides.

#### 2.3.2. Fumigant Toxicity against Adults

In previous studies, 1,8-Cineole has been proven to perform better as a fumigant when compared with his capacity as a contact acaricide [43,50]. For the non-contact toxicity of the volatile active compound, a bean leaf disk was placed inside a 4 cm Petri dish. Using a stereomicroscope, 20 adult female mites, between 3 and 5 days of age, were carefully placed on the leaf disk using a small tip painting brush, the Petri dish was then covered with filter paper to allow gas exchange and the dish was then placed inside a 9 mm Petri dish (Figure 1). Active compound (1,8-Cineole and its nanoemulsion) was applied to a 1 × 1 cm square of filter paper attached to the bottom surface of the 9 cm Petri dish lid, at concentrations ranged from 0 to 90 mg/L air in 10 mg/L intervals. Concentrations mg/L air represent the amount of substance per volume of air in Petri dish. Distilled water was used as negative control for evaluation of free monoterpene, and a mixture of water and surfactants in the same proportion as the emulsion was used as negative control for the evaluation of the nanoemulsion. Petri dishes were left unsealed to allow the monoterpene to volatilize in non-controlled conditions. As positive control for this comparative study, free 1,8-Cineole was used in sealed dishes to evaluate its toxicity without volatilization. For this and every other bioassay, three independent replicates were carried out, each one of which was performed by triplicate. Dead mites were counted under stereomicroscope after 24 h of treatment [43]. Mites were considered dead if appendages did not exhibit any movement after being prodded with a fine brush.

### 2.4. Aphicidal Activity

#### 2.4.1. Tested Animals

Tested aphids *R. maidis* Fitch were reared on corn (*Zea mays* L.) tassels in the Laboratory of Natural Products Chemistry, Faculty of Chemistry and Engineering, Autonomous University of Baja California. Rearing conditions were 26 ± 2 °C and 70 ± 5% R.H. with a 16:8 h (L:D) natural photoperiod. Aphids were not previously exposed to any pesticides.

#### 2.4.2. Fumigant Toxicity against Wingless Adults

For the non-contact toxicity of the volatile active compound, the same arrange described for acaricidal assays in Section 2.3.2 was used with modifications: for this assay, 20 wingless adult aphids, between 3 and 6 days of age, were placed on a corn leaf disk and 1,8-Cineole and its nanoemulsion were evaluated at concentrations ranged from 0 to 120 mg/L air in 20 mg/L intervals. Dead aphids were counted under stereomicroscope after 48 h of treatment [58]. Distilled water was used as negative control; for effectivity comparison, 1,8-Cineole in sealed conditions was also evaluated. Aphids were considered dead if appendages did not exhibit any movement after being prodded with a fine brush.

### 2.5. Insecticidal Activity

#### 2.5.1. Tested Animals

Tested insects *B. tabaci* Gennadius were reared on pumpkin (*Cucurbita maxima* L.) leaves in the Laboratory of Natural Products Chemistry, Faculty of Chemistry and Engineering, Autonomous University of Baja California. Rearing conditions were 26 ± 2 °C and 70 ± 5% R.H. with a 16:8 h (L:D) natural photoperiod. Silverleaf whiteflies were not previously exposed to any pesticides.

#### 2.5.2. Fumigant Toxicity against Adults

For the non-contact toxicity of the volatile active compound, a similar experiment as the previously described in Section 2.3.2 was carried out. An 8 cm disc of pumpkin leaf was placed on the bottom of a 9 cm Petri dish. An amount of 20–30 adult silverleaf whiteflies, between 2 and 4 days old, were collected with an adapted transfer pipette, and placed on the pumpkin leaf and covered with a nylon mesh [59]. 1,8-Cineole/nanoemulsion was applied to a 1 × 1 cm square of filter paper attached to the bottom surface of the Petri dish lid, at concentrations ranging from 0 to 100 mg/L air in 20 mg/L intervals. Distilled water was used as negative control. Dead whiteflies were counted under stereomicroscope after 9 h of treatment [60]. Whiteflies were considered dead if appendages did not exhibit any movement after being prodded with a fine brush.

### 2.6. Data Analysis

Mortality percentages were obtained and corrected according to Abbott’s formula [61]. Lethal concentrations of LC_50_ and LC_95_ were estimated using log-dose PROBIT analysis [62] (IBM SPSS statistics 22, Chicago, IL, USA). For each experiment, three repetitions were carried out.

## 3. Results and Discussion

### 3.1. Characterization of Nanoemulsion

#### 3.1.1. Visual Appearance and Droplet Size

The appearance of the nanoemulsion to the naked eye, 24 h after synthesis, was highly transparent with a slight blue color (Figure 2). Nanoemulsion did not exhibit any phase separation or visible particles floating.

Droplet size of 1,8-cineole nanoemulsion was 14.73 ± 0.203 nm, representing the diameter that corresponds to the maximum intensity of Figure 3, and a PDI of 0.178 ± 0.01, meaning nanoemulsion was monodisperse. While particle size in nanoemulsions usually ranges between 20 and 200 nm [63], for monoterpenes nanoemulsions droplet sizes between 17 and 19 nm have been reported using the Phase Inversion Method and as small as 7 nm using a sonicator [64,65].

#### 3.1.2. Evaluation of Stability of the Nanoemulsion

Formulation, stored at room temperature, was exposed to centrifugation to determine its stability against gravitational force. Centrifugation can accelerate destabilization mechanisms that can occur at a slower pace, including sedimentation or flocculation. Nanoemulsions are considered stable if after centrifugation test, they exhibit no signs of separation or aspect modifications, which was the case for the prepared 1,8-cineole nanoemulsion. The additional emulsion with oil:surfactant:water ratio of 0.5:0.5:9 did not pass the centrifugation test, showing clear sings of emulsion rupture, and was not analyzed on Turbiscan. These results are consistent with previous reports that showed an increased surfactant concentration can result in a more stable emulsion [65,66].

#### 3.1.3. Analysis of Stability of the Nanoemulsion with Turbiscan LAB^®^

To study destabilization mechanisms that can result in an emulsion rupture, nanoemulsion was analyzed on Turbiscan LAB^®^, which provides a real-time prediction of the most common mechanisms such as creaming, flocculation, coalescence, and sedimentation. Measurement of stability is based on the variation on the transmission and backscattering signals as a result of migration or coalescence of droplets. The sample (24 h old nanoemulsion) was analyzed without dilution or further preparation. Delta transmission and delta backscattering profiles during 1 h scan (Figure 4A,B) were studied to determine possible instability patterns in the formulation.

Transmission signals were uniform during the entire scan, with variations of ±0.3%, throughout the entire analysis, with no detected destabilization phenomena occurring in the sample.

Backscattering can yield information on homogeneity of particle distribution: an increase in backscattering at the bottom of the sample can translate in migration of particles from top to bottom (sedimentation), while an increase in backscattering at the top of the sample can be a result of creaming.

Delta backscattering of the nanoemulsion was within the interval ±1%, indicating no migration, coalescence, or flocculation in the sample during the entire analysis, as well as no variation in particle size, suggesting that the nanoemulsion is highly stable over time [67,68].

Any variation in both backscattering and transmission profiles under 2.5 mm and over 39 mm, visible in Figure 4A,B, are not a result of destabilization process, but are related to distortions in the bottom of the glass vial and the meniscus and air on the top.

### 3.2. Evaluation of Sealed 1, 8-Cineole against Arthropod Pests T. urticae, and R. maidis

#### 3.2.1. Acaricidal Activity in Petri Dishes

The acute toxicity of vapors of 1,8-Cineole in three different conditions—sealed dish, free and in a nanoemulsion—were evaluated against the spider mite *Tetranychus urticae,* the toxicity parameters for each of the treatments is displayed on Table 1. After 24 h treatment, as expected, 1,8-Cineole in sealed conditions exhibited the highest acaricidal activity with 98% mortality at 30 mg/L air (Figure 5), and an LC_50_ of 19.5 mg/L air, and LC_95_ of 32.34 mg/L air, this results exhibited a weaker fumigant activity than that reported by Badawy et al. (2010) of 4.80 mg/L air and by Abdelgaleil et al. (2019) with LC_50_ 6.67 mg/L air [43,50]. Additionally, this results are in accordance to fumigant bioactivity of 1,8-Cineole on other stored food mites that has also been reported, with Sánchez-Ramos and Castañera (2000) reporting more than 90% mortality on mite *Tyrophagous putriscentiae* at a 1,8-Cineole concentration of 66.7 µL/L and an LC_50_ of 14.9 µL/L [42], and it was reported as less effective on mite *Tyrophagus longior*, causing only 50% mortality at a similar concentration [69].

#### 3.2.2. Aphicidal Activity in Petri Dishes

Vapors of 1,8-Cineole exhibited aphicidal activity against *R. maidis* in all three treatments. Cineole in sealed dishes had the highest bioactivity after 48 h exposure, with an LC_50_ of 31.98 mg/L air and an LC_95_ of 59.18 mg/L air. Concentration was higher than that obtained on the acaricidal assay, suggesting a lower aphicidal activity in comparison; however, 100% mortality was reach at 80 mg/L air (Figure 6) and all live aphids appear lethargic for several minutes after exposure, which could be further studied.

To our knowledge, there are not previous studies on 1,8-Cineole toxicity on *Rhopalosiphum maidis;* nevertheless, there are some reports on 1,8-Cineole containing essential oils activity against this aphid, with Benddine et al. (2022) reporting acute toxicity of essential oil of *Myrtus communis*, 28% of its composition being 1,8-Cineole with corrected mortality rates of 88.13% and 81.47% at concentrations of 6ml/L and 5ml/L, respectively, after a 5-day exposure [70]. Essential oil of *Tanacetum vulgare* (16.8% 1,8-Cineole) also exhibited with insecticidal activity towards the green peach aphid *Myzus persicae* (Sulzer) [71]. The toxicity parameter for all three treatments in this assay are shown in Table 2.

#### 3.2.3. Insecticidal Activity against *B. tabaci*

Toxicity against silverleaf whitefly *B. tabaci* was only evaluated for free 1,8-Cineole and the nanoemulsion (Figure 7). Sealed conditions seemed to not be tolerated by this insect, with all individuals’ dead after 9 h treatment, including the negative controls. Free 1,8-Cineole reached the highest toxicity level of all three bioassays, with a LC_50_ of 61.3 mg/L air (Table 3). Susceptibility of *B. tabaci* to volatile compounds in essential oils has been stablished by several studies, with essential oils of *Thymus vulgaris* at concentrations of 0.5% causing mortality up to 79% on different developmental stages of *B. tabaci* [72]. Liu et al. (2014) reported strong fumigant toxicity of 16 essential oils against *Bemisia tabaci* with LC_50_ in the range from 0.11µg/L to 13.54 µg/L [73].

Insecticidal activity of 1,8-Cineole on diverse crop and stored food pests has been widely studied. Liu et al. (2021) reported acute lethality of vapors of 1,8-Cineole on third instar larvae of *Spodoptera litura*, obtained a LC_50_ of 7 µL/L air after 24 h treatment [74]. 1,8-Cineole is the main constituent of the EO of *Lavandula dentata*, with reported insecticidal activities on *Sitophilus zeamais*, *Tribolium castaneum* and *Epicauta atomaria*, with LC_50_ concentration ranging from 11.3 to 26.9 µL/L air [75].

While these concentrations are notoriously lower than the ones been reported in this study, is important to highlight that the arrangement of this in vitro assay was designed to mimic volatilization of the monoterpene under non-controlled conditions to establish the potential enhanced activity with nanoformulations, which will be discussed in Section 3.2.4.

#### 3.2.4. Evaluation of Free Compound and Nanoemulsion Three against Arthropod Pests

Since open field would highly differ from sealed Petri dish conditions, the focus of this discussion is on the comparison between the toxicity exhibited by pure, free 1,8-Cineole and the nanoemulsion. Free monoterpene registered a notoriously lower toxicity than its sealed counterpart against *T. urticae* and *R. maidis* requiring, respectively, near to 7 and 4 times more than the concentration required by the sealed compound to achieve 50% mortality; this indicates that a large part of the monoterpene is lost due to volatilization or degradation, diminishing its bioactivity. Free compounds exhibited LC_50_ of 137.46 mg/L air, for acaricidal activity, while nanoemulsion required 55.23 mg/L air to achieve the same mortality, meaning that 1,8-cineole nanoemulsion was near to 2.5 times more effective than the free monoterpene. For aphicidal activity, LC_50_ for the free compound was observed at 140.27 mg/L air, meanwhile, the LC_50_ for the nanoemulsion was 73.05, almost twice the biocidal activity was observed for the nanosystem. For insecticidal activity, LC_50_ of 61.308 and 30.98 mg/L air were observed for the free monoterpene and the nanoemulsion, respectively; however, statistical analysis proved there was not significant differences between both concentrations, when the corrected mortality percentages (Figure 7) are compared, as an interest contrast between the two systems is observed at the lower concentrations, i.e., until 60 mg/L air, nanoemulsion caused more than twice the mortality of silverleaf whiteflies than the free monoterpene, suggesting that the free compound requires a higher dosage to inflict acute toxicity on *B. tabaci* while a nanosystem seems to lower the concentration required.

While nanoemulsion did not reach the level of acute toxicity of the sealed compound, which could correspond to a slower release due to retention of the compound from the micelles, in free, environment-exposed conditions, this same characteristic can provide the volatile terpene of improved chemical stability, which is translated in an enhanced bioactivity when compare to free compound, the nanosystem seemed to promote a paulatine release of the compound, prolonging the exposure, which results in a virtually higher toxicity when compared to the free monoterpene. Similar findings have been reported for other manoterpene and essential oil nanoemulsions against a variety of arthropod pests. Almadiy (2021) [76] reported a superior bioactivity of a nanoemulsion of *Achillea biebersteinii* on the red flour beetle when compared with the free essential oil and its main monoterpenes; likewise, Suresh et al. (2020) found that the sea fennel in nanoemulsion was close to twice as bioactive than its free form [77]. In the same way, various essential oil nanoemulsions have demonstrated to be more effective against agriculture and stored food pests than the free essential oil, this is appreciated in a lower concentration requirement to neutralized 50% (LC_50_) of arthropod population in toxicity assays [78,79,80].

Of all three biocidal activities evaluated, *B. tabaci* was the most susceptible to 1,8-Cineole in its free and nanoemulsion forms, requiring the lowest concentration to reach 50% mortality on the population, as well as requiring less exposure time, followed by *T. urticae*, and the monoterpene was the least effective on *R. maidis*, requiring the longest exposure time and the highest concentration of free and compound in a nanoemulsion.

## 4. Conclusions

A nanoemulsion of 1,8-Cineole with an oil:surfactant:water ratio of 0.5:1:8.5 was formulated by the low energy method of EPI, and a mean droplet size of 13.4 nm was obtained. Nanoemulsion high stability was determined by centrifugation and Turbiscan analysis, with delta backscattering shifts within the interval ±1%, indicating no migration, coalescence or flocculation in the emulsion. Fumigant application of 1,8-Cineole exhibited higher acute toxicity against adult two-spotted spider mites *T. urticae,* corn leaf aphid *R. maidis* and whitefly *B. tabaci,* when compared to the effects of the free monoterpene, reducing LC_50_ parameter in more than 50% for all three pests.

The results obtained in this comparative study suggest that nanoemulsions can provide bioactive volatile compounds, such as monoterpenes and essential oils, for protection from rapid volatilization, allowing adequate exposure time and doses, which results for enhanced activity. The higher rates of mortality of all three pests proposes this nanoemulsion as a potential botanical nano-pesticide product against agriculture arthropod pests. Further studies are necessary to evaluate efficiency of this nanoformulation on open field and/or greenhouse conditions.

## Figures and Tables

**Figure 1 insects-14-00663-f001:**
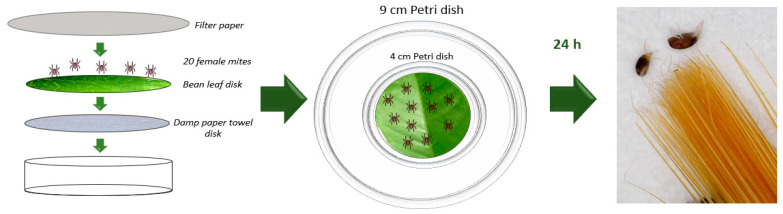
Fumigant toxicity assay arrangement.

**Figure 2 insects-14-00663-f002:**
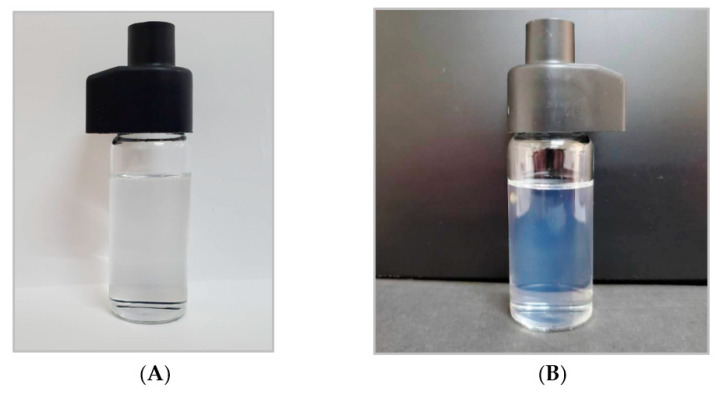
(**A**) 1,8-Cineole nanoemulsion against white surface. Nanoemulsion did not exhibit any separation or disturbance to the naked eye and showed a high level of transparency. (**B**) Natural light passing through the nanoemulsion, showing a transparent blue color.

**Figure 3 insects-14-00663-f003:**
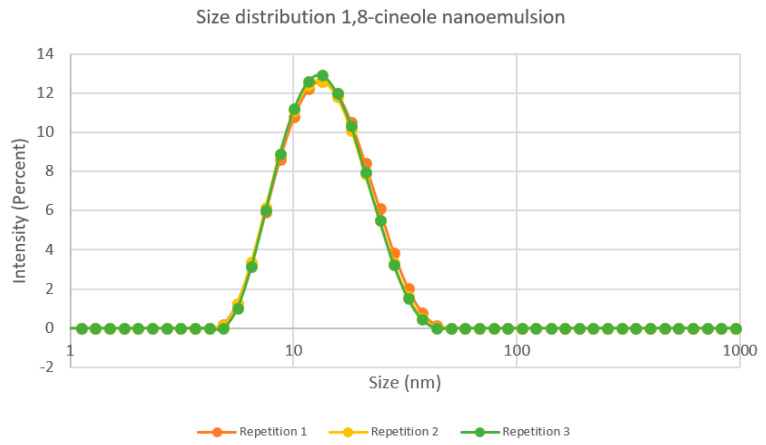
Size distribution of the diameter of the particles (in nm) by intensity of the nanoemulsion of 1,8-cineole. Data taken from triplicates of one of three independent experiments.

**Figure 4 insects-14-00663-f004:**
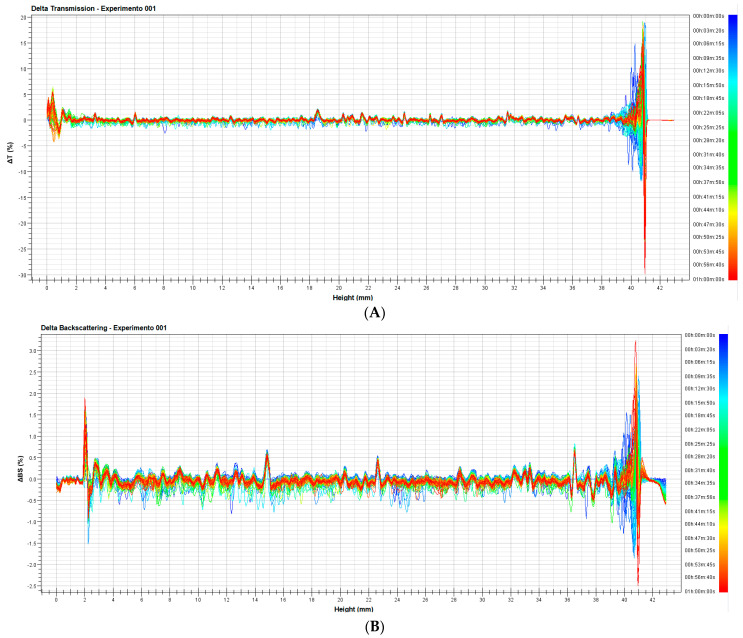
(**A**) Delta transmission profile and (**B**) Delta backscattering profile of the nanoemulsion. Data are reported as a function of time (1 h) and sample height (mm).

**Figure 5 insects-14-00663-f005:**
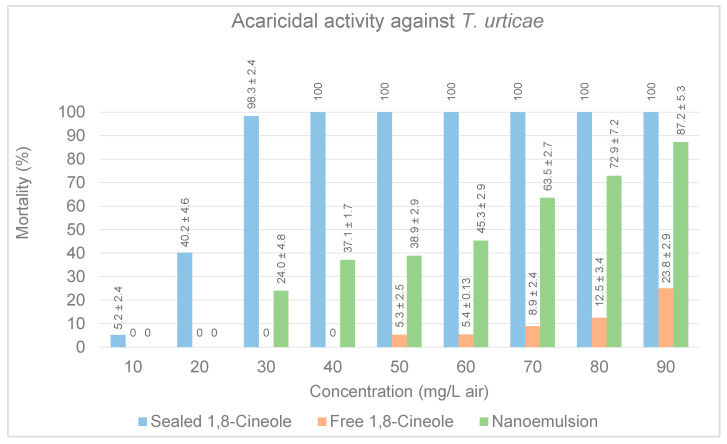
Comparative effects of sealed, free and nanoemulsion conditions of 1,8-Cineole on mortality rates of female populations of *T. urticae.* Means ± standard deviation of corrected mortality.

**Figure 6 insects-14-00663-f006:**
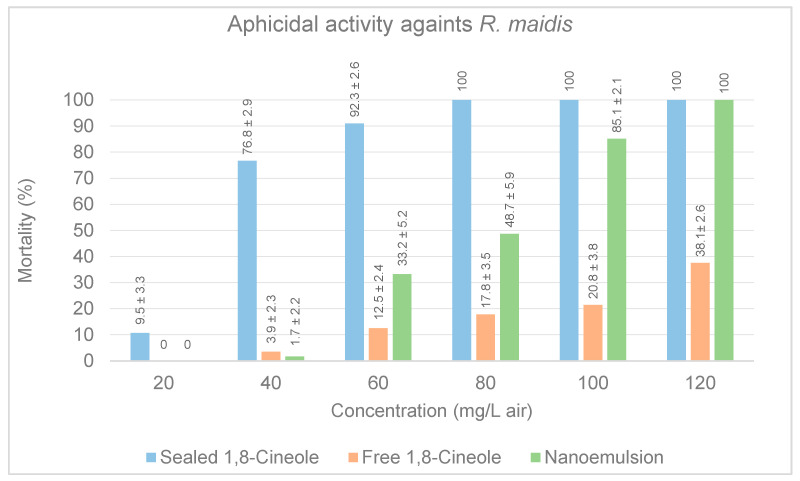
Comparative effects of sealed, free and nanoemulsion conditions of 1,8-Cineole on mortality rates of populations of *R. maidis.* Means ± standard deviation of corrected mortality.

**Figure 7 insects-14-00663-f007:**
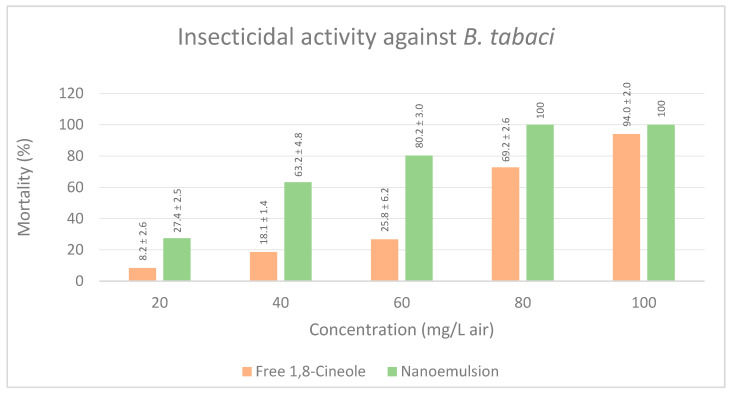
Comparative effects of free 1,8-Cineole, and its nanoemulsion on mortality rates of populations of *B. tabaci.* Means ± standard deviation of corrected mortality.

**Table 1 insects-14-00663-t001:** Fumigant activity of 1,8-cineole in sealed dishes, free, and its nanoemulsion on mortality of female spider mites, *T. urticae*.

1,8-Cineole	N ^1^	LC_50_ (CI) ^2^ mg/L Air	LC_95_ (CI) ^2^ mg/L Air	Slope ± SE	Χ^2^(df)	*p*-Value
**Sealed**	1060	19.5 (16.43–22.70) a ^3^	32.34 (26.72–50.14) a	7.484 (0.387)	40.62 (4)	0.000
**Free**	1080	137.46 (106.89–317.92) c	329.49 (189.43–2283.43) bc	4.333 (0.593)	8.67 (4)	0.070
**Nanoemulsión**	1440	55.23 (48.16–63.59) b	140.90 (108.11–233.64) b	4.044 (0.218)	38.65 (6)	0.000

^1^ Total number of spider mite adult females assayed. ^2^ LC_50_ and LC_95_ represent the concentrations mg/L air (in Petri Dish) required to kill 50% and 95% of spider mite adult females, respectively; 95% confidence intervals (CI) are shown in parentheses. ^3^ Different letters in columns indicate significant differences in LC50 and LC95 values.

**Table 2 insects-14-00663-t002:** Fumigant activity of 1,8-cineole in sealed dishes, free and its nanoemulsion on mortality of *R. maidis* aphid.

1,8-Cineole	N ^1^	LC_50_ (CI) ^2^ mg/L Air	LC_95_ (CI) ^2^ mg/L Air	Slope ± SE	X^2^(df)	*p*-Value
Sealed	720	31.98 (23.89–39.53) a ^3^	59.18 (46.56–101.24) a	6.152 (0.367)	7.09 (2)	0.029
Free	900	140.27 (118.44–207.68) c	240.36 (185.12–430.57) bc	0.016 (0.002)	6.659 (3)	0.086
Nanoemulsion	720	73.05 (58.96–86.50) b	118.01 (96.53–212.25) ab	7.897 (0.445)	28.88 (3)	0.000

^1^ Total number of spider mite adult females assayed. ^2^ LC_50_ and LC_95_ represent the concentrations mg/L air (in Petri Dish) required to kill 50% and 95% of spider mite adult females, respectively; 95% confidence intervals (CI) are shown in parentheses. ^3^ Different letters in columns indicate significant differences in LC50 and LC95 values.

**Table 3 insects-14-00663-t003:** Fumigant activity of free 1,8-cineole and in nanoemulsion on mortality on adults of *B. tabaci*.

1,8-Cineole (Sealed)	N ^1^	LC_50_ (CI) ^2^ mg/L Air	LC_95_ (CI) ^2^ mg/L Air	Slope ± SE	X^2^(df)	*p*-Value
Free	1200	61.308 (33.34–100.32) ab ^3^	152.93 (95.97–4689.22) ab	4.144 (0.239)	96.42 (4)	0.000
Nanoemulsion	1350	30.98 (21.69–38.51) a	80.44 (62.43–131.25) a	3.969 (0.201)	33.93 (4)	0.000

^1^ Total number of spider mite adult females assayed. ^2^ LC_50_ and LC_95_ represent the concentrations mg/L air (in Petri Dish) required to kill 50% and 95% of spider mite adult females, respectively; 95% confidence intervals (CI) are shown in parentheses. ^3^ Different letters in columns indicate significant differences in LC50 and LC95 values.

## Data Availability

The data presented in this study are available in the article.

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
