# Peer review of "Evaluation of the Stability of a 1,8-Cineole Nanoemulsion and Its Fumigant Toxicity Effect against the Pests Tetranychus urticae, Rhopalosiphum maidis and Bemisia tabaci"

_insects, 2023, doi:10.3390/insects14070663_

Round 1
Reviewer 1 Report
Line 94-95. No need to mention full form of the pest species again.
Line 131. 26 ± 131 2°? temperature? add the unit please. Also 16:8 h.
Author Response
Dear Reviewer,
We thank you for your time and effort in reviewing our manuscript. The feedback has been invaluable in improving the content and presentation of the paper.
We have revised our manuscript according to your comments.
We hope the revised paper is suitable for inclusion in the journal Insects.
Yours sincerely,
Joaquín González Marrero.
Response:
- Line 94-95. No need to mention full form of the pest species again.
Response: Corrected.
- Line 131. 26 ± 131 2°? temperature? add the unit please. Also 16:8 h.
Response: Units were added
Reviewer 2 Report
Dear authours
This manuscript is unpublished and has the potential to be published. This study aimed to evaluate the toxicity of 1,8-cineole nanoemulsion for important agricultural pests: Tetranychus urticae, Rhopalosiphum maidis, and Bemisia tabaci. The authors compared the toxicity of the 1,8-cineole nanoemulsion with the unformulated compound; the nanoemulsion proved stable. Some considerations were made in the text, mainly regarding the methodology to specify some points. Improvements should be made regarding the discussion. The authors need to develop some aspects of the discussion. Reports on the pesticidal activity and mode of action of 1,8-cineole for the arthropods studied in this work may be scarce however, this is a well-studied secondary metabolite. Thus, the authors can use studies conducted with other arthropods for the discussion, mainly regarding hypothesizing the mode of action. Could this compound be acting on the nervous system of these arthropods? It is not explored in the discussion, from the point of view of arthropod physiology, why a substance in a nanosystem can be more toxic?
Best regards
Line 4: Del point after Bemisia tabaci.
Line 35: First citation of pests, therefore scientific names should not be abbreviated
Line 42: Insert terms for indexing (Keywords) that facilitate the article's citation and do not appear in the title and abstract, for example, IUPAC name of 1,8-cineole; spider mite etc...
Line 75: chance to “the indiscriminate and abusive use of synthetic chemical insecticides to benefit the selection of resistant populations”
Line 91: You should better justify the advantage of a nanoemulsion
Line 93: State the hypothesis of your work. Would the nanoemulsion be more efficient?
Lines 107, 110 and 111: change - Triton X-100 and Tween 20 (7:3) for ( g:g or mL/mL); 5% oil (w/w) for g; 85% water (w/w) for mL.
Lines 118 to 120: add in the subitem 2.1
Line 115: prior to analysis, the sample was solubilized... what concentration?
Line 122: Stable formulations?? Was more than one formulation made? If so, it must be described. You just described it in the results.
Line 129: T. urticae
Line 132: Aphids or mites???
Line 132: The creation was maintained according to conditions already established in the literature?? Which? Authors used as references must be cited.
Line 139: What is the approximate age difference between the females? The approximate variation in age between insects must be described for all tests.
Line 160: According to some method described in the literature?
Line 179: B. tabaci – the scientific name must be abbreviated after the first citation in the text
Line 226: could have add this information in the methodology
Lines 266 to 278: the important thing is to compare 1,8-Cineole in its free form and in nanoemulsion; so the presentation of results would be better this way: toxicity of 1,8-Cineole nanoemulsion was x times more higher when ....
Fig 6. For all figures, the presentation of data would be better visualized in a table, given that in some cases there was no mortality in some treatments (negative control)
Lines 305 to 313: the same comments above are valid
Line 366 to 395: Ok, but the authors need to discuss why this toxicity is higher for the compounds in the nanosystem. The discussion is too superficial
Line 395: can it be considered an encapsulated compound?
Author Response
Dear Reviewer,
We thank you for your time and effort in reviewing our manuscript. The feedback has been invaluable in improving the content and presentation of the paper.
We have revised our manuscript according to your comments.
We hope the revised paper is suitable for inclusion in the journal Insects.
Yours sincerely,
Joaquín González Marrero.

Reviewer 3 Report
It is an interesting manuscript, because it provides bases for the control of three important species of pests worldwide. The development of insecticides of natural origin is a topic of great interest within the framework of sustainable agriculture. However, there are aspects that need to be reviewed and clarified.
Line 17 . It says: "... natural products is an effective alternative". This is a strong statement in my opinion. Consider your review.
Line 21, line 49, line 64,... and others. Whitefly is the English name of the species of the superfamily Aleyrodoidea. The English name of Bemisia tabaci is tobacco whitefly.
Line 42. keywords: eucalyptol, Rhopalosiphum maidis and Bemisia tabaci should be added
Line 56. It says: "Rhopalosiphum maidis Fitch (Homoptera: Aphididae), ..." . It should say: "Rhopalosiphum maidis Fitch (Hemiptera: Aphididae)"
Line 89 to 92. "However, in realistic settings, EOs and monoterpenes are not as effective due to their high volatility, low water solubility, and chemical instability. Nanocarriers are proposed as a potential solution that allows a practical application of these compounds". I think this needs a citation.
Line 132. It says: "Aphids were not..." . It should say: "Acari were not..."
Line 143. Specify that the concentrations refer to the active substance per volume of air in the Petri dish.
Line 149 It says: "Three replicates were carried out for each concentration." However, in line 285 (also 323, and 358) states "Three replicates were performed for each three independent experiments." It is not clearly explained how many replicates and repeats were performed. I deduce that three independent experiments (= replicates) are carried out. And in each of them 3 repetitions or repeats are made (3 Petri dishes) by concentration. It should be better explained.
Lines 197 to 200. Data analysis. The statistical analysis of the calculation of the lethal concentration is poorly specified. What software is used? How have you dealt with 0% or 100% mortality. The statistical treatment of the data is not clear and I believe that it can clearly be improved.
Line 207. "Droplet size of 1,8-cineole nanoemulsion was 14.73 ± 0.203 nm". You must specify what this value represents. Is Z-average = Particle mean diameter in nm?. Is the diameter that corresponds to the maximum intensity of figure 3?
Line 218. It says "Fig. 3. Size distribution of nanoemulsion...". It should say: "Size distribution of the diameter of the particles (in nm) by intensity of the nanoemulsion..."
Line 226. It says "An additional emulsion with an oil: surfactant ratio of 0.5:0.5 was formulated ...·" You should specify the proportion of water.
Line 265. It says "3.2.1 Acaricidal activity in sealed Petri dishes". It should say: 3.2.1 "Acaricidal activity in Petri dishes"
Line 282. It says " Means ± standard deviation". It should say "Mean ± standard deviation of corrected mortality".
Tables 1, 2 and 3. The value indicated as chi-square is the p-value, but the values of the statistic and the degrees of freedom are not indicated. And LCL and UCL are Lower Confidence Limit and Upper Confidence Limit, respectively.
Line 293. It says "3.2.2 Aphicidal activity in sealed Petri dishes". It should say: 3.2.2 "Aphicidal activity in Petri dishes"
Line 296. The indicated values do not coincide with Table 2.
Lines 322, 366, 396 The header numbering is incorrect.
Line 334-336. In figure 7 does not show the treatment of "Sealed conditions". I don't understand what "negative controls at 80 ppm" means.
Line 560. The original paper is from 1925 and can be found at https://doi.org/10.1093/jee/18.2.265a
Check the scientific names of the species, there are many that are not written in italics, also in the bibliography
Best regards
Author Response

(The authors gave the same response as above.)
